# Postoperative Activity and Knee Function of Patients after Total Knee Arthroplasty: A Sensor-Based Monitoring Study

**DOI:** 10.3390/jpm13121628

**Published:** 2023-11-21

**Authors:** Sebastian Kersten, Robert Prill, Hassan Tarek Hakam, Hannes Hofmann, Mahmut Enes Kayaalp, Jan Reichmann, Roland Becker

**Affiliations:** 1Center of Orthopaedics and Traumatology, University Hospital Brandenburg/Havel, Brandenburg Medical School Theodor Fontane, 14770 Brandenburg an der Havel, Germany; 2Department of Orthopaedic Surgery, Sana Kliniken Sommerfeld, 16766 Sommerfeld, Germany; 3Faculty of Health Sciences Brandenburg, Brandenburg Medical School Theodor Fontane, 14770 Brandenburg an der Havel, Germany; 4Istanbul Kartal Research and Training Hospital, 34865 Istanbul, Turkey; 5StatConsult GmbH, 39112 Magdeburg, Germany

**Keywords:** inertial measurement unit, TKA, remote monitoring, knee function, rehabilitation

## Abstract

Inertial measurement units (IMUs) are increasingly being used to assess knee function. The aim of the study was to record patients’ activity levels and to detect new parameters for knee function in the early postoperative phase after TKA. Twenty patients (n = 20) were prospectively enrolled. Two sensors were attached to the affected leg. The data were recorded from the first day after TKA until discharge. Algorithms were developed for detecting steps, range of motion, horizontal, sitting and standing postures, as well as physical therapy. The mean number of steps increased from day 1 to discharge from 117.4 (SD ± 110.5) to 858.7 (SD ± 320.1), respectively. Patients’ percentage of immobilization during daytime (6 a.m. to 8 p.m.) was 91.2% on day one and still 69.9% on the last day. Patients received daily continuous passive motion therapy (CPM) for a mean of 36.4 min (SD ± 8.2). The mean angular velocity at day 1 was 12.2 degrees per second (SD ± 4.4) and increased to 28.7 (SD ± 16.4) at discharge. This study shows that IMUs monitor patients’ activity postoperatively well, and a wide range of interindividual motion patterns was observed. These sensors may allow the adjustment of physical exercise programs according to the patient’s individual needs.

## 1. Introduction

There is a growing interest in the use of sensors in orthopaedics. Sensor-assisted Total Knee Arthroplasty or Sensor-guided Arthroscopy Training are common examples [1,2].

Wearable inertial measurement units (IMUs) are commonly used for assessing lower extremity kinematics or after knee surgery [3]. They have gained popularity for monitoring gait and joint function after TKA [4,5]. These systems allow a more objective assessment of joint function offering detailed information about patients’ activity and, thus, allowing the adjustment of postoperative rehabilitation programs on a more individual basis [6]. A large number of different sensor types are used, depending on the aim of the investigations with the result of a wide variety of spatiotemporal gait parameters [3]. No standardization in measurement techniques or test methodologies has been established so far [4]. Sensors with triaxial accelerometers are most frequently used in the functional assessment of patients. Other uses include either inertial sensors equipped with triaxial accelerometers, gyroscopes or magnetometers [4,6]. In addition, there is a significant variance in the sampling rate of data recordings and the application of the sensor to the patient [4]. Accelerometers allow extended monitoring of the general activity of patients due to better battery life and low power consumption. The limitation of accelerometers is the lack of data quality referred to as complex gait parameters in comparison to IMUs. Most accelerator systems were attached to the trunk, at the hip or back. Knee joint-specific parameters have not been primarily investigated so far [4]. IMUs are valid and reliable for TKA patients.

IMUs enable the acquisition of more complex gait parameters and activity patterns. Previously published studies analyzed the pre- and postoperative activity level measured by the patient’s maximum knee angle, the impact load, the asymmetry in the impact load, the knee joint flexion while walking, the number of steps taken and the time spent sitting. The most commonly used measurement of patients’ activity level is the number of steps taken and range of knee motion [4,7,8,9,10,11,12]. However, current IMUs are unable to clearly differentiate between certain activities of daily life (ADL), such as walking, sitting, standing, lying down, climbing stairs, getting passive motion or physical therapy [7]. Most sensor-based examinations after knee replacement refer to the functional analysis preoperatively and weeks to months postoperatively. As the research group showed in previous work, sensor-based assessment is reliable, but the functional assessment of improvement after TKA and also in hip arthroplasty is diverse, not consistent and lacks adherence to core outcome sets [6,13,14,15,16]. Most sensor-based examinations after knee replacement refer to the functional analysis preoperatively and weeks to months postoperatively. It is known that early and standardized recovery concepts after TKA and total hip arthroplasty (THA) significantly reduce the length of hospital stay and the occurrence of complications [17,18]. However, the function of the knee joint and activity of the patient in the early postoperative phase after TKA lacks objective data.

The aim of this study was to record the postoperative level of patients’ activity in more detail during their hospital stay after primary TKA using a novel IMU. Therefore, the main purpose of the study is to prove the feasibility of providing complex and novel movement parameters during a hospital stay. Novel parameters were measured analyzing knee function and patient’s activity after surgery better. It was hypothesized that the novel IMU would allow the characterization of different patient activities in more detail quantitatively during the early postoperative period.

## 2. Materials and Methods

### 2.1. Participants

From September 2021 to March 2022, a total of 20 patients (n = 20) were prospectively enrolled in the study.

The inclusion criterion for this study comprised patients treated with primary total knee arthroplasty. Patients with unicondylar knee arthroplasty; postoperative complications, such as bleeding, signs of infection, severe joint effusion; an allergy to plasters, cognitive deficits or multimorbidity (ASA III–IV), were excluded from the study.

Ethical approval was obtained from the Ethics Committee of the Landesärztekammer Brandenburg, Germany (ref.-number: AS 92(bB)/2018). All participants gave written consent.

### 2.2. Sensors

The Orthronic Smart Knee (OSK) sensor works with an inertial measurement unit, including a 3-axis accelerometer/gyroscope/magnetometer. Other integrated sensors record air pressure and temperature. The size of the sensor was 37 mm × 26 mm × 8 mm (L × W × H) and a total weight of 6 g. The sensors recorded knee motion at a sampling rate of 25 Hz and transmitted it to a tablet via Bluetooth. The sampling rate for measuring the air pressure, temperature and the number of steps was reduced to 1 Hz.

If the IMU was unable to transfer data directly via Bluetooth, measurement data were automatically saved in the integrated 512 MB memory in offline mode.

In the present study, the 290 mAh lithium polymer battery enabled measurement intervals of 24 h in total.

The reliability of the sensors was proven during development by comparison to a 10-camera motion capture system (VICON-MX-S, Vicon Motion System Ltd., Oxford, UK) with defined movements. A deviation with a mean of 5 degrees was shown [19]. Other patient-related factors that could affect the reliability and accuracy of the sensors, such as the attachment to the skin or the tilting of the sensors due to muscle movements, were also examined using the VICON^®^ camera system. The more complex motion analysis of the sensors showed a deviation of a maximum of six degrees.

### 2.3. Sensor Application

The measurements started on the first postoperative day. The proximal and distal sensors were positioned on the ventral distal thigh over the quadriceps tendon and the medial tibia (Figure 1). After connecting the sensors with the mobile application via Bluetooth, the data were continuously recorded in offline mode for 24 h a day. The duration of recording the motion was up to six days, from the first postoperative day to when the patients were discharged (POD 1–6) from the hospital. The sensors were changed daily every afternoon. For documentation purposes, a picture of the patient’s knee with the applied sensor was taken every time. Additionally, patients protocoled their daily activities, such as walking, going to the toilet, different exercises, physical therapy and climbing stairs, including the corresponding time in which these activities took place. After the sensors were removed, data were extracted to a tablet via Bluetooth.

### 2.4. Measurements

#### 2.4.1. Daily Distribution of Standing, Horizontal and Sitting Position

The integrated accelerators allowed the analysis of patients’ positions. A standing position was characterized by the vertical orientation of the thigh, a sitting position by the horizontal orientation of the thigh and the vertical orientation of the lower leg and a horizontal position by the horizontal orientation of both the upper and the lower part of the leg (Figure 2). To ensure that sensor vibrations did not affect the recording of the posture, measurements made using the accelerometer were filtered based on average data. The size of the data window was 25 samples, which corresponds to a data collection time of one second.

The definition of a standing position was the inclination of the upper leg with more than 45 degrees with respect to the transverse plane (α > 45°). The definition of a sitting position was the combination of an inclination of the upper leg with less than 45 degrees and an inclination of the lower leg with more than 30 degrees with respect to the transverse plane each (α < 45°, β > 30°). The lying position was defined as a combination of an inclination of the upper leg with less than 12 degrees and an inclination of the lower leg with less than 30 degrees with respect to the transverse plane each (α < 12°, β < 30°). The data were analyzed for daytime from 6 a.m. to 8 p.m.

The postures were calculated at each measurement time. After that, all samples at which a standing, sitting or lying posture was detected were counted. The total time for each posture was computed relating to the number of samples meeting the described angle criteria.

#### 2.4.2. Movements

A detected movement of the knee joint is defined as a significant change in knee angle and direction of movement (Figure 3). The threshold was set for a significant change in knee angle to 10 degrees to distinguish clearly between real movements and tremors or flickers. In addition, it was not assumed that the knee angle would return to the initial value after the movement. An example of that is the transition from a horizontal to a sitting position. A new movement was detected as soon as the knee angle moved more than 10 degrees from the last maximum and ended when a counter movement of more than 10 degrees was detected.

#### 2.4.3. Steps

Algorithms for counting steps, which are standard in many accelerators, are supposed to be less sensitive at the early stage after surgery because of the slow walking speed and limited weight bearing.

Patient step counting was calculated based on the detected posture and changes in the knee angle. A standing posture with a significant change of the knee angle of more than 20 degrees was counted as a step (Figure 4).

#### 2.4.4. Movement over Angular Velocity

The knee angle is measured at a rate of 25 times per second by the OSK-Sensor. The angular velocity can be calculated using the first-time derivative.
$$Angular\;velocity=\frac{change\;of\;knee\;angle}{time\;interval}$$

In the present measurements, the time interval was 40 ms.

#### 2.4.5. Continuous Passive Motion (CPM)

CPM was defined as a significant change in knee movements with an angle change of more than 10 degrees in a supine position of the patient. Furthermore, the movements were slow and lasted longer than 12 s but less than a minute. A CPM training session involved at least 10 resembling motions within a 15 min interval.

A minimum of 20 repetitions were executed during a session.

#### 2.4.6. Stationary Cycling (MOTOmed^®^)

During the hospital stay, patients performed training sessions on a stationary bike (MOTOmed^®^, RECK-Technik GmbH & Co. KG, 88422 Betzenweiler, Germany). This exercise was performed while being seated and took less than three seconds per repetition. During practice, these repetitive motions took place in rapid succession. If at least 250 of these repetitive motions were detected without a break, the period was classified as a MOTOmed^®^ exercise session.

### 2.5. Statistical Analysis

All patient and operation-related data were pseudonymized and transferred to an Excel sheet using the serial number “S00XX”. The patient’s names and case numbers were removed. Assigning case-related data to specific patients was thereby not possible. All continuous variables were expressed as mean and standard deviation. Figures have been prepared with Python for SPSS V26 (IBM SPSS Inc., Chicago, IL, USA).

No inductive statistics were used as no statistical hypothesis had to be proven. Analysis was performed on a visual and documentation basis.

## 3. Results

### 3.1. Demographics

A total of 20 patients participated in the study, including nine males (n = 9) and eleven females (n = 11), with a mean age of 69.6 years (SD ± 8.8) and a mean BMI of 29.9 (SD ± 5.6). The demographic data of the participants are shown in Table 1.

### 3.2. Measurements of Patient Activity and Knee Function

#### 3.2.1. Steps per Hour

The mean number of steps on the first day after surgery was 117.4 (SD ± 110.5) and increased to 858.7 steps (SD ± 320.1) at the time of discharge (POD 6). The average activity level of patient S0011 is shown in Figure 5. The activity level was reflected by the number of steps taken per hour. Step frequency increased significantly from the first postoperative day.

The average number of steps performed by the patients during the entire measurement period between day 1 and day 6 was 526.2 steps per day (SD ± 431.9). The mean knee flexion angle of all patients during walking was 30.9° across all days (SD ± 6.5). The mean walking time of the patients during the entire measurement period was 16.1 min per day (SD ± 11.1).

#### 3.2.2. Analysis of Postures during Daily Activity

There was an increase in the general activity with an increasing time of standing and sitting positions and a decrease in the time of horizontal position. Figure 6 shows the results of patient S0017 for illustrative purposes. The distribution of the position of the patients on the first day after surgery was 91.2% in a horizontal position, 6.4% in a sitting position and 2.4% in a standing position and changed to 69.9% in a horizontal position, 22.9% in a sitting position and 7.1% in a standing position at the final day.

#### 3.2.3. Physical Therapy, CPM-Device

CPM was performed for a mean of 36.4 min per day (SD ± 8.2). An example of a detected CPM training is shown in Figure 7. The mean knee angle achieved was 82.9 degrees (SD ± 21.7) with a repetition rate of 42.1 (SD ± 16.3) movements per application. Ninety degrees of knee flexion was achieved between the third and fourth day after surgery (Figure 8).

#### 3.2.4. Knee Flexion over Angular Velocity

Data evaluation of angular velocities used in motion revealed a significant trend over the period of early postoperative rehabilitation. The mean velocity for a defined knee joint movement with an angle change of more than 10 degrees regardless the activity was 12.2 degree per second on day one of the measurement (SD ± 4.4). On the last day of the measurement, the mean velocity was 28.7 degrees per second (SD ± 16.4). Patients moved their affected knee joints in a small range and at low speeds in the first few days after surgery. During their hospital stay, they increased their range of motion and angular velocity, which could already be seen at the time of discharge from the hospital. This is shown in Figure 9 and Figure 10 using patients S0011 and S0017 as an example.

#### 3.2.5. Analysis of Stair Climbing

Going up and down stairs is one of the patient’s most important therapeutic goals for the activities of daily life. This is characterized by a rapid change in altitude detected by the air pressure sensor with simultaneous knee activity, as shown in Figure 11. Using an elevator is an alternative to climbing stairs in the early period after surgery. The change in altitude is also recognized by air pressure, but the associated knee movement is missing (Figure 12).

## 4. Discussion

The most important finding of the study is that IMUs offered detailed monitoring of different patient’s activities after TKA in the early postoperative period. The hypothesis, that the novel IMU allows to characterize different patient’s activities in more detail quantitatively at the early postoperative period, can be accepted. The recording of several functional parameters of the knee joint made it possible to compare the activity level of the participants with the result of major differences in the level of postoperative mobilization between patients.

In addition to recording functional parameters of the knee joint, the sensors allowed to identify various activities of the patient’s daily life. Patient’s therapy goals during the hospital stay and criteria of discharge can be objectified quantitatively with the help of data analysis of the IMU. The sensors could be an important tool to analyze the postoperative progress in patient mobilization and recovery in real time. Due to the more detailed assessment of patients, activity rehabilitation programs can be individualized according to patients’ need.

The integration of different sensor types into the IMU allowed us to distinguish different functional and activity parameters, which could be quantified using algorithms. However, some of the parameters had to be read out manually, such as climbing stairs or using an elevator. Different methods are used in sensor-based measurements of knee patients. The review of Small et al. has shown that the type of sensors being used for gait analysis changed between 2008 and 2016. Initially, uniaxial accelerometers were used. Recent publications mostly imply triaxial IMUs [1].

A two-sensor-based measurement method was used in the current study, with one sensor placed over the anterior distal femur and the other over the proximal medial tibial surface. A similar position was reported by others [9]. However, alternatively, sensor positioning may also be in the sagittal plane at the lateral femur and tibia for instance [20]. There are also measuring systems with multiple sensors available, which are attached in different positions on the body and often fixed with multiple belts, such as in the DynaPort^®^-System [21,22]. It seems that none of the sensor systems allow a consistently high level of accuracy in all motion variables. It depends on which parameters are examined [23]. The usage of more than two sensors makes recording daily activity rather impossible. The key determinant of applicability seems to be the simple usage for patients.

The threshold of 20 degrees was chosen to distinguish true steps with near-normal knee function from movements rather than performed with a stiff knee joint while walking. Thus, a change of knee angle of less than 20 degrees was not detected routinely by the current algorithm. Therefore, the number of steps may be underrepresented. The threshold is still of debate and may be reduced to 15 degrees for instance. Hayashi et al. used an IMU attached with a belt around the waist and showed a significantly higher rate of counted steps per day for patients in the early postoperative phase after TKA (POD 3–10) [24]. However, the IMU attached to the waist might also have detected other motion activities than the overall step counting. The number of steps is recorded by others using a single IMU according to an algorithm, which distinguished between static and dynamic activities, counting all peaks in the vertical signal as steps and all peaks in the anteroposterior signal as a complete gait cycle (2 steps) [25].

In order to record the different postures of the patients, we used an algorithm that calculated the position of the two sensors in relation to one another. Based on previously defined angles, we differentiated between a sitting, standing or horizontal position. Lipperts et al. chose another method by using the inclination angle of a single triaxial accelerometer for distinguishing between sitting and standing positions and defined them as static activities [25].

An increase in the physical activity of the patients was observed during early rehabilitation at the hospital. While patients predominantly were lying in bed during the first few days, the activity pattern shifted to a higher percentage of standing and sitting positions. However, significant heterogeneity between the patients was noted.

Patients’ motivation but also patients’ preparation for surgery and the early postsurgical period showed a significant impact [17,18]. Taking the current data into consideration, it shows that early inpatient mobilization after TKA was rather insufficient. There seems to be a lack of encouragement for patients’ mobilization. The results emphasize the need for advanced recovery concepts. Generally, perioperative care and integrated education programs are insufficiently focused but hold high potential for outcome improvement [26].

The movements in regard to angular velocities showed an interesting trend over the entire time of recording. Already in the first days after surgery, the patients used higher knee angles with a higher speed more frequently. An explanation might include the reduction in pain and swelling, improved coordination while walking and a wider range of motion. Muscle function may improve at the early phase as well, which requires a more detailed investigation of muscle function prior to and at the early stage after surgery.

The level of overall activity during hospital the stay was rather low. These data cannot be generalized, however, as they represent the results of a single center. These data are valuable to improve patients’ care during hospitalization and can be used for quality assessment purposes.

The collected data showed that patients presented a very individual level of mobility. In some cases, the percentage between the standing and supine positions differed in an inversely proportional manner to the number of days after surgery. Some patients had a larger number of steps taken on the 2nd than on the 3rd or 4th postoperative day. Nevertheless, it should be noted that the recording period involved the weekends as well. Less activity during the weekend in general seems to be partially caused by fewer physiotherapy sessions. It also showed that the patient’s mobility seems to rely mainly on daily physiotherapy compared to self-mobilization, which is partially in line with other findings [27] but supports the need for defining precise programs for home-based approaches. This also shows the strong need for digital and virtual reality rehabilitation approaches [28,29,30]. The importance of compliance issues is still unknown when self-mobilization is recommended. A high variance in the level of activity of patients after TKA was reported by others, but it has to be mentioned that the measurement with a watch is a limiting factor in the comparison of the data with our study [31].

The sensors are not only applicable for general monitoring of patient’s daily activity but may also be used while performing specific exercises. The Sensor and accompanying System will be used in an already planned clinical trial on Compliance parameters for physical therapy interventions after Total Knee Arthroplasty with a focus on APP-based rehabilitation. It might also be used in combination with Neural Networks for decision making in primary or revision TKA [32]. They could also further inform predictive model studies on pain or length of stay [33].

Due to the ease of use and independence from the clinical environment, the IMU could also measure patients’ mobilization after discharge from the hospital providing important information to the Orthopedic Surgeon and practitioner at the follow-up examination.

## 5. Conclusions

The objective parameter of movements over angular velocities could be a new decisive criterion for the patient’s knee function. The wearable IMUs are a quick and easy solution to measure a patient’s activity level and could contribute to low-threshold monitoring of postoperative knee function to help quantify improvements in both activity and knee motion following TKA. The usage of IMUs might aid in the development of home-based physical exercise programs. These programs might be adjusted to the individual needs of patients.

## 6. Limitation

A limiting confounder could be the visible presence of the sensors, which could have led to an increase in compliance by both the patients and the physical therapists. Furthermore, some of the participants in the study were on different wards with different therapists, who might have administered different rehabilitative exercises. On the other hand, this often reflects the reality of inpatient physiotherapeutic care. The number of steps might have been underrepresented due to our algorithm.

Additionally, the length of inpatient stays was different in some cases, which had an impact on the duration of the postoperative measurement and, thus, on the increase in the activity pattern.

## Figures and Tables

**Figure 1 jpm-13-01628-f001:**
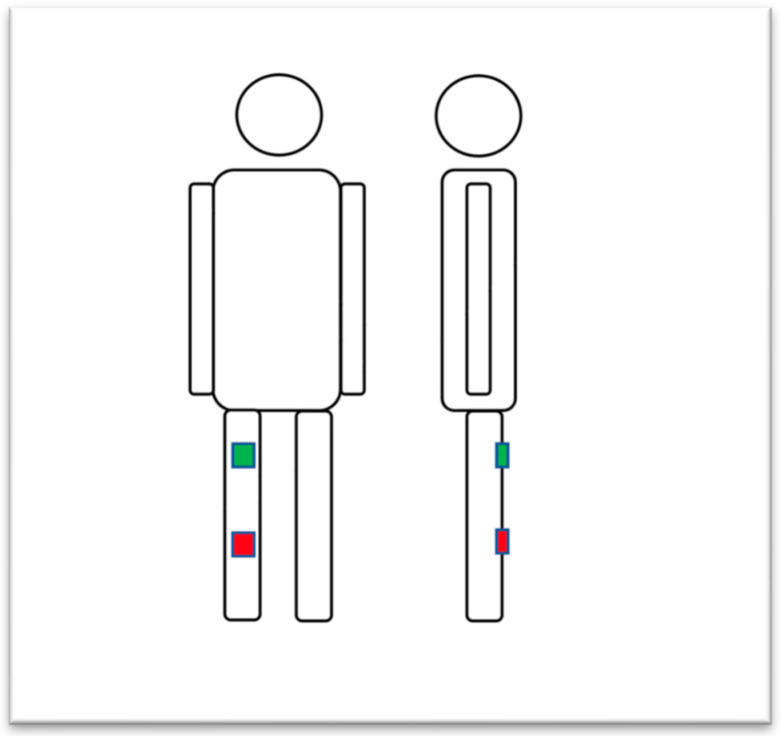
Sensor placement at the femur and tibia using sticky tape.

**Figure 2 jpm-13-01628-f002:**
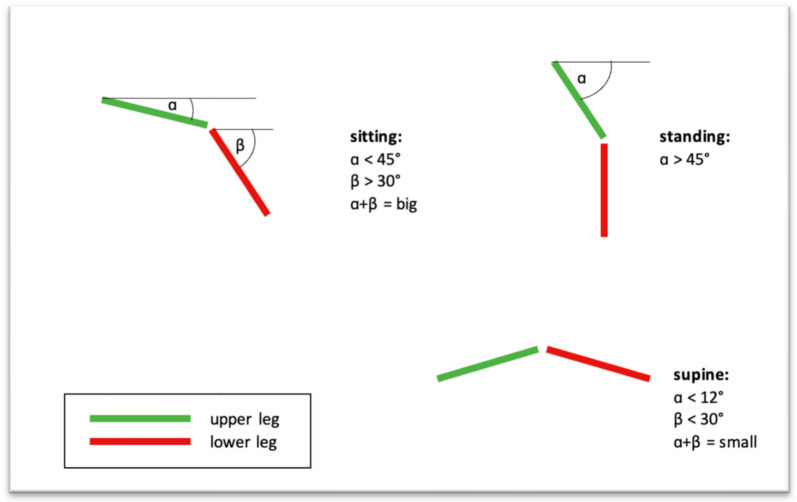
Algorithms for detecting different postures of the patients, such as sitting, standing and lay position by analyzing the position of the femur and tibia.

**Figure 3 jpm-13-01628-f003:**
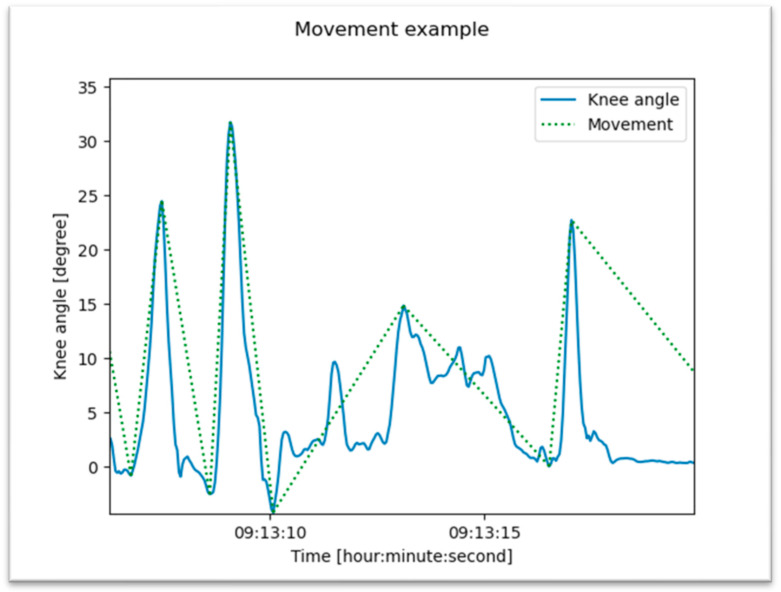
Algorithm to detect and discriminate real movements from trembling or flickering. A significant change in flexion angle greater than 10 degrees (threshold 10°) was counted as a movement.

**Figure 4 jpm-13-01628-f004:**
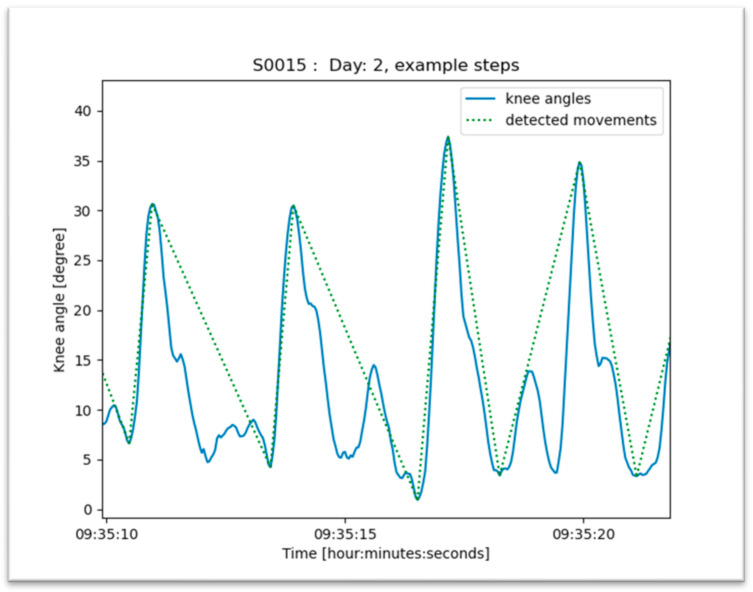
Algorithm to detect steps. Steps were detected when a standing posture with a significant change of the knee angle greater than 20 degrees was measured by IMUs (threshold > 20 degrees).

**Figure 5 jpm-13-01628-f005:**
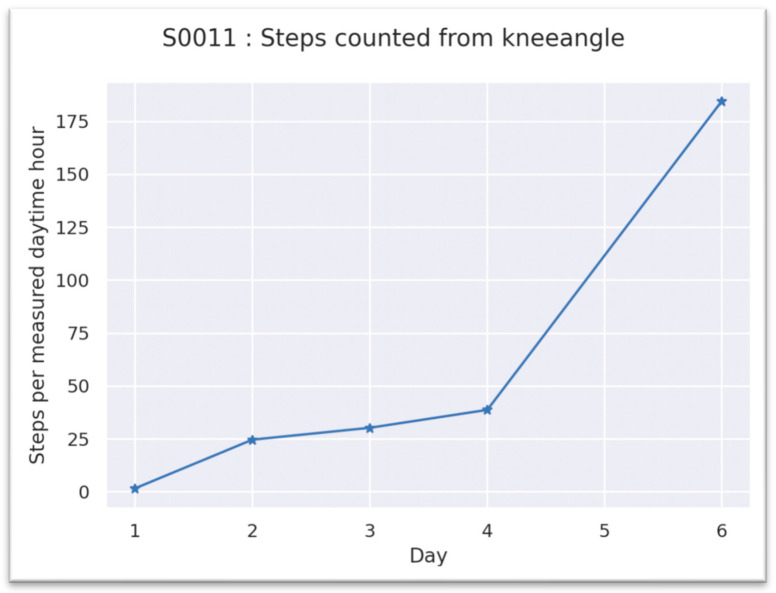
Steps normalized per hour. Day 1 = first day after surgery, Patient S0011.

**Figure 6 jpm-13-01628-f006:**
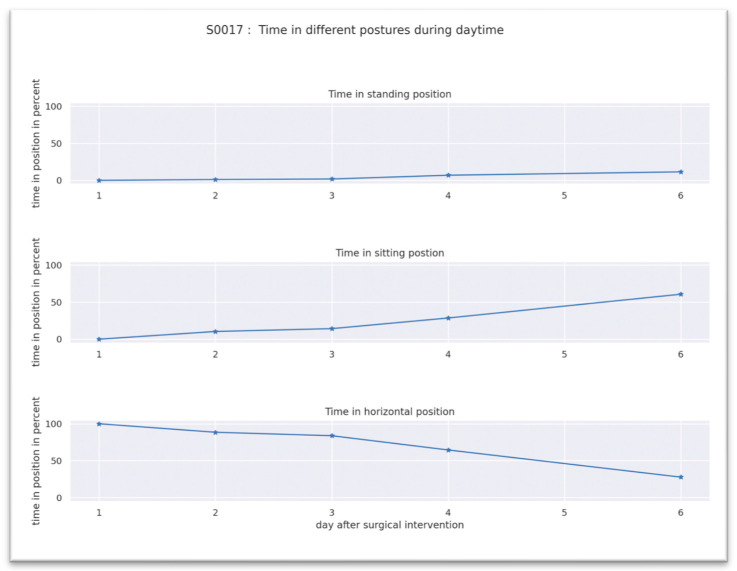
Time given in percentage of different postures from 6 a.m. to 8 p.m. of Patient S0017.

**Figure 7 jpm-13-01628-f007:**
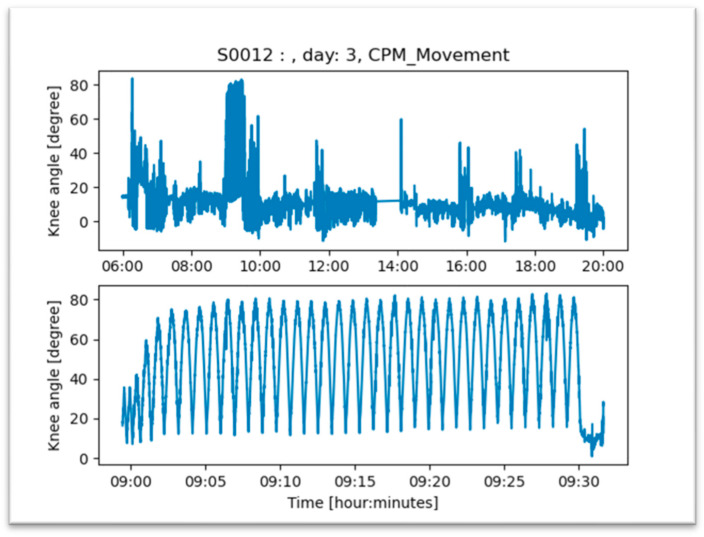
Example of a detected CPM training session for patient S0012 during 9.00 and 9.30 am. Despite the fact that the patients can move the knee up to 80 degrees, active movement was significantly lower throughout the day.

**Figure 8 jpm-13-01628-f008:**
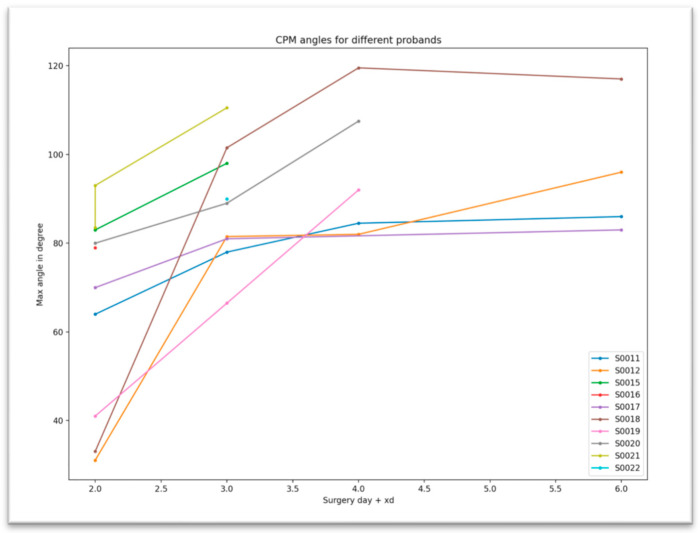
CPM-detected continuous passive motion angles over days.

**Figure 9 jpm-13-01628-f009:**
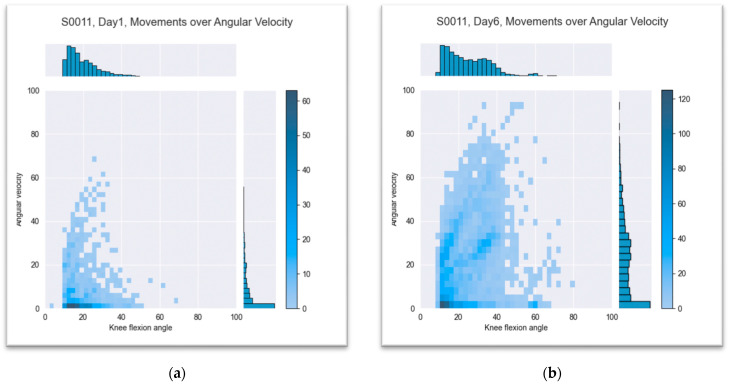
Knee flexion over angular velocity for patient S0011: (**a**) first day after surgery and (**b**) last day of measurement, sixth postoperative day. The color of the squares shows the distribution of the flexion angle in regard to the angle velocity. The dark color means more frequent movement.

**Figure 10 jpm-13-01628-f010:**
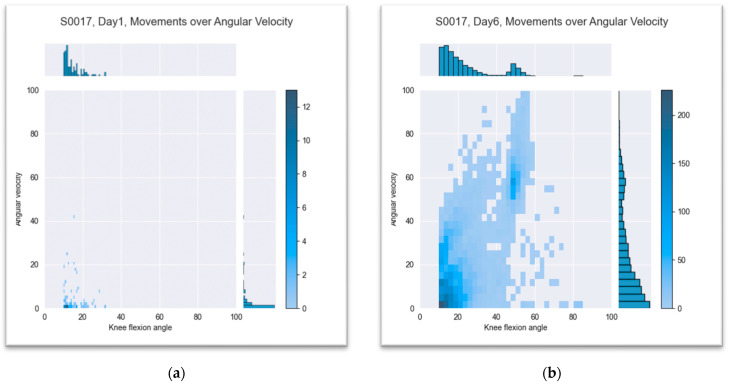
Knee flexion over angular velocity for patient S0017: (**a**) first day after surgery and (**b**) last day of measurement, sixth postoperative day. The color of the squares shows the distribution of the flexion angle in regard to the angle velocity. The dark color means more frequent movement.

**Figure 11 jpm-13-01628-f011:**
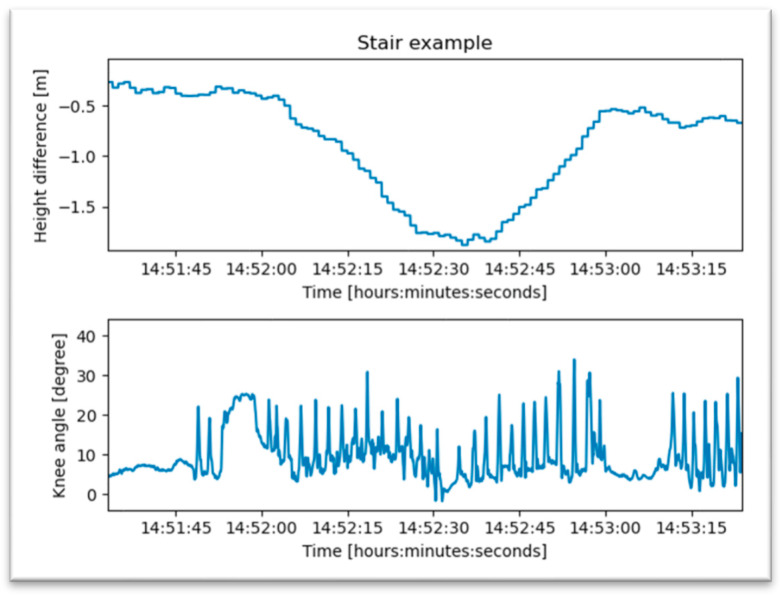
Example of a patient who is climbing stairs.

**Figure 12 jpm-13-01628-f012:**
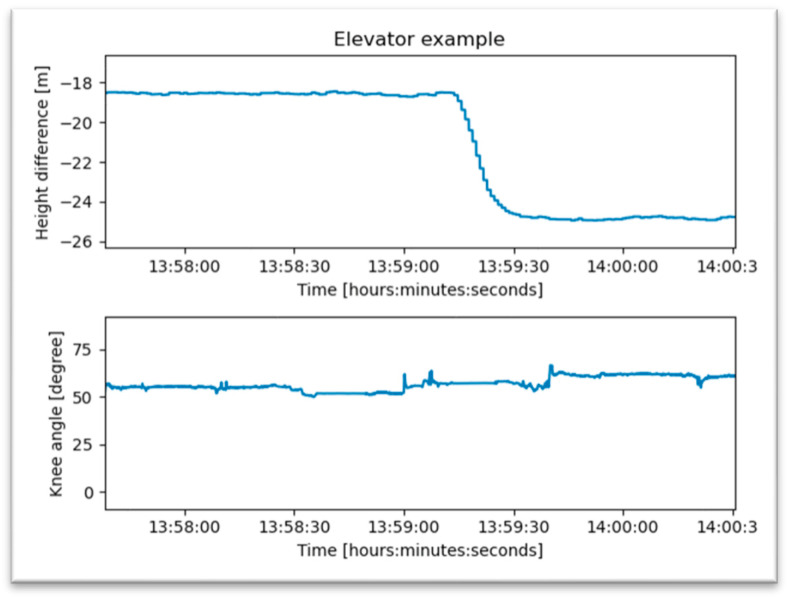
A patient using an elevator.

**Table 1 jpm-13-01628-t001:** Patients’ demographic data.

Patient Data	Overall (n = 20)
female: male (n)	11: 9
Age (years)	69.6 (SD ± 8.8)
Heigth (in m)	1.7 (SD ± 0.1)
Weight (in kg)	84.4 (SD ± 19)
BMI (kg/m^2^)	29.9 (SD ± 5.6)

All continuous variables were expressed as mean and standard deviation. Abbreviations: BMI—Body Mass Index.

## Data Availability

Additional Data is available on request from the corresponding author.

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
