# Peer review of "Postoperative Activity and Knee Function of Patients after Total Knee Arthroplasty: A Sensor-Based Monitoring Study"

_jpm, 2023, doi:10.3390/jpm13121628_

Round 1

Reviewer 1 Report

Comments and Suggestions for Authors

the paper is dealing with a very interesting topic

But the authors did not put a clear aim to the study, is it the sensitivity of the sensors, or the compliance of patients, or just to say this method is useful and could help in collecting data 

I think the authors need a clear main aim of the study and according to this the methodology and interpretation can be adjusted 

Author Response

Reviewer 1.

Dear Reviewer, thank you so much for your review of our article Postoperative Activity and Knee Function of Patients after Total Knee Arthroplasty: A Sensor-based Monitoring Study

Ths helped us to further improve.

The paper is dealing with a very interesting topic

Thank you very much for your comment.

But the authors did not put a clear aim to the study, is it the sensitivity of the sensors, or the compliance of patients, or just to say this method is useful and could help in collecting data .I think the authors need a clear main aim of the study and according to this the methodology and interpretation can be adjusted 

Thank you very much. We added a more precise aim and revised accordingly in the discussion the main aim should now be obvious. Changes are marked in yellow.

Once again, thank you so much for your help with paper.

Reviewer 2 Report

Comments and Suggestions for Authors

1. Please check references, typos, etc. according to the journal format.

2.  You must fill in the references in lines 32-33. Please provide references for all sentences presented.

3. The paragraphs in the introduction are too detailed. Please combine paragraphs on the same topic.

4. Can you do sample size calculation?

3. Were they evaluated by raters blinded to the study?

4. Please fill out the reliability and validity of the evaluation tool.

5. Please provide a reference for the IMU.

6. Please add more clinical significance to the discussion of this article.

7. Please provide references to the sentences presented in the discussion.

Strengths: This is a study that allows detailed evaluation of the movement of Total Knee Arthroplasty by applying IMU.

Weakness: The disadvantage is that it cannot be evaluated in all clinical locations

Comments on the Quality of English Language

English language is fine.

 Please check references, typos, etc. according to the journal format.

Author Response

Reviewer 2

Dear Reviewer, thank you so much for your review of our article Postoperative Activity and Knee Function of Patients after Total Knee Arthroplasty: A Sensor-based Monitoring Study

This helped us to further improve.

Please check references, typos, etc. according to the journal format.

Thank you very much. We revised all references according to MDPI style and revised also typos and punctation.

You must fill in the references in lines 32-33. Please provide references for all sentences presented.

We added the requested references.

The paragraphs in the introduction are too detailed. Please combine paragraphs on the same topic.

Thank you very much. We agree and combined paragaphs where possible.

Can you do sample size calculation?

Unfortunately, this is not possible due to the nature of the study. It is exploratory and you can not estimated any effects or precission in advance. Fortunately, this paper will now help to calculate sample size for upcoming studies.

Were they evaluated by raters blinded to the study? Please fill out the reliability and validity of the evaluation tool.

Thank you for the question. I combine the answer as both address the same issue. This is a study were the analyst is provided with data to develop the algorithm. A reliability study on the algorithm could be a next step. Therefore, blinding is not feasible.

Please provide a reference for the IMU.
The reference for the older version of the IMU is in the text. The article for the  validation for the new version is under preparation.

Please add more clinical significance to the discussion of this articlePlease provide references to the sentences presented in the discussion.

Thank you. We added the upcoming projects to the discussion, which highlights the the clinical relevance. But we did not cite the protocol or compliance paper to not further increase self citation.

Strengths: This is a study that allows detailed evaluation of the movement of Total Knee Arthroplasty by applying IMU.

Thank you very much!

Weakness: The disadvantage is that it cannot be evaluated in all clinical locations

Once the system is available on the market, we hope it will be evaluated in many settings as really belief there is value for clinical evaluations.

Please find all changes in the manuscript in track change or highlighted in yellow.

Once again, thank you so much for your time and help with the paper.

Round 2

Reviewer 1 Report

Comments and Suggestions for Authors

the paper now is suitable for publication